# Polymer-Embedding Germanium Nanostrip Waveguide of High Polarization Extinction

**DOI:** 10.3390/polym15204093

**Published:** 2023-10-15

**Authors:** Jinyuan Liu, Ziyang Zhang

**Affiliations:** 1College of Information Science and Electronic Engineering, Zhejiang University, Hangzhou 310027, China; liujinyuan@westlake.edu.cn; 2Laboratory of Photonic Integration, School of Engineering, Westlake University, Hangzhou 310024, China

**Keywords:** polymer waveguide technology, integrated optics, germanium waveguide, spectroscopic ellipsometry, hybrid integration

## Abstract

Germanium (Ge) nanostrip was embedded in a polymer and studied as a waveguide. The measurements reveal that this new type of semiconductor/polymer heterogeneous waveguide exhibits strong absorption for the TE mode from 1500 nm to 2004 nm, while the propagation loss for the TM mode declines from 20.56 dB/cm at 1500 nm to 4.89 dB/cm at 2004 nm. The transmission characteristics serve as an essential tool for verifying the optical parameters (*n*-*κ*, refractive index, and extinction coefficient) of the strip, addressing the ambiguity raised by spectroscopic ellipsometry regarding highly absorbing materials. Furthermore, the observed strong absorption for the TE mode at 2004 nm is well beyond the cut-off wavelength of the crystalline bulk Ge (~1850 nm at room temperature). This redshift is modeled to manifest the narrowing of the Tauc-fitted bandgap due to the grain order effect in the amorphous Ge layer. The accurate measurement of the nanometer-scale light-absorbing strips in a waveguide form is a crucial step toward the accurate design of integrated photonic devices that utilize such components.

## 1. Introduction

Germanium (Ge), as one of the first-generation semiconductor materials [1], has been widely used in various semiconductor devices and integrated circuits (ICs) [2,3,4]. Thanks to its narrow bandgap and high electron mobility upon doping [5], Ge has become a key material for the development of high-speed photodetectors at near-infrared (NIR) wavelengths [6,7,8,9]. Recently, Ge thin films have been coated on fiber facets and have shown excellent saturable absorption effects in fiber lasers, making this traditional semiconductor material an attractive candidate for the development of novel mode-locked lasers [10,11,12].

Over the years, many Ge-based micro/nano optical waveguides have been reported [13,14,15,16,17,18,19]. These waveguides usually exhibit strong absorption in the O, C, and L bands [20] and mainly operate at mid-infrared (MIR) above 2 μm [15,16,17,18]. Such waveguides typically adopt a channel or ridge structure using crystalline Ge films with a thickness on the micrometer scale [19]. The epitaxy/chemical vapor deposition for such thick Ge layers can be critical, and the cost of the Ge-on-insulator (GeOI) wafers remains high. A simple and low-cost way to fabricate a Ge-based waveguide is highly desired for the development of compact and advanced photonic devices.

On the other hand, polymer waveguide technology has matured and become an excellent platform for high-performance photonic devices [21,22,23], thanks not only to the degree of freedom in molecular engineering and chemical process control [24,25] but also the compatibility to integrate components from other material platforms [26,27]. This flexible feature allows other inorganic materials to be embedded in the polymer as the key optically functional layer, e.g., metals, dielectrics, and semiconductor thin films. Moreover, the fabrication technology for polymer waveguide devices is relatively simple, and the optical loss can be suppressed at customized wavelength windows. The polymer material exhibits good thermal tunability [21] and has hence led to a series of novel functional devices, including tuneable external-cavity lasers [28], optical phased arrays [29], thermo-optic switches [30], and chip-level optical computing devices [31].

Furthermore, recent results have shown that polymer waveguides made of perfluorinated acrylate resins can transmit high-peak-power femtosecond laser pulses with low loss, good stability, and linearity [32]. As polymer waveguides have already been used to build continuous-wave (CW) external-cavity lasers [28], further integration of waveguide components with saturable absorption ability may lead to the development of low-cost and high-performance mode-locked lasers on the chip scale. Therefore, the investigation of Ge nanostrips as waveguides in polymer becomes an essential step toward this goal.

In previous works, researchers have tried to integrate 2D material such as graphene into the polymer waveguide platform [33,34]. However, the technology to transfer graphene is challenging and often results in large areas of defects, whereas Ge can be easily deposited on wafers using standard e-beam evaporation. On the other hand, regardless of integrating novel 2D materials or conventional thin-film materials, an accurate characterization of their optical parameters is an inevitable step for the precise design of the waveguide component. Though low-defect 2D materials can be deposited by advanced CVD (chemical vapor deposition) technology [35] and the film quality becomes good enough for reliable ellipsometric measurement, the subsequent modeling of the optical parameters (*n*-*κ*) can still lead to ambiguous results [36,37]. Hindered by the numerical, trial-and-error nature of the modeling process, it is difficult to pin down the optical properties of thin films with strong absorption/light–matter interaction in the chosen wavelength range. Basically, the modeling process “infers” the extinction coefficient *κ* from Im (*ε*) (the imaginary part of the complex dielectric function) with a series of hypothetical oscillators, the type, number, and sequence of which do not need to follow a fixed, physical pattern, so long as the combination minimizes the fitting error. Therefore, it could lead to multiple solutions but also bring non-negligible errors near the absorption cut-off wavelength. To be able to adopt the nanometre-scale Ge film as the key functional layer in photonic devices, e.g., as a saturable absorber in an on-chip mode-lock laser, it is necessary to find an additional tool to verify and correct its optical parameters, complementary to ellipsometry.

To address these problems, in this work, we investigated the feasibility of embedding a thin Ge nanostrip in polymer to form a waveguide using conventional fabrication technology. The waveguide loss is characterized rigorously by the cut-back method using three different waveguide lengths at well-controlled TE (transverse electric) and TM (transverse magnetic) states. The waveguide near-field profile emitting from the facet is also captured using a well-referenced imaging system to calculate the mode field size. Then, a *n*-*κ* model of the amorphous Ge thin film is measured and modeled by a spectrometric ellipsometer. Comparisons and corrections to the *n*-*κ* model are made based on the waveguide characterization, e.g., the transmission loss and the mode field size. Moreover, the corrected *κ* model can well-explain the redshift of the cut-off absorption wavelength in the amorphous Ge film compared to the single-crystalline bulk. This work may serve as the foundation to further exploit Ge as well as other semiconducting and metal films for advanced functions on the polymer waveguide platform, e.g., as a polarization-selective absorber, saturable absorber, dispersion, and nonlinear optical component, etc.

## 2. Material and Methods

### 2.1. Waveguide Design

The schematic view of the germanium nanostrip waveguide is shown in Figure 1a. A standard silicon wafer is used as the carrier only, without any optical or electrical functions. A high-aspect-ratio (~100) Ge nanostrip is embedded in a commercial perfluorinated polymer (Exguide ZPU12-450, ChemOptics, Daejeon, Republic of Korea) cladding with a refractive index of 1.45 at a 1550 nm wavelength. In this highly customized resin, the C-H bonds in the polymeric chain are mostly replaced by C-F bonds for a lower light absorption at infrared wavelengths compared to the unoptimized acrylates and some inorganic optical materials [22,23]. This polymer material also functions as an electronic blocking layer to embed the Ge nanostrip without noticeable ion diffusion. Figure 1b illustrates the normalized mode profiles of the TE and TM fundamental modes of the Ge waveguide at a 1550 nm wavelength, respectively. Also shown are the complex refractive indexes and mode field areas (MFAs). This highly birefringent property is typical when the core has a large aspect ratio. Some examples include thin silicon nitride waveguides in glass [38] and polymer [39].

For the TE mode, the dominating field component is E_y_, and it concentrates in the Ge core, which results in strong absorption in the NIR. On the other hand, for the TM mode, the dominating field component is E_z_, and it expands more in the customized low-loss polymer cladding and hence may lead to a lower absorption than Ge waveguides on conventional material platforms. In Figure 1b,c, the modes are propagated over a distance of 50 μm and plotted under the same color bar for comparison. The complex refractive index of bulk single-crystalline Ge is taken in this simulation as a starting point. At NIR wavelengths, the designed Ge nanostrip waveguide is multimode for the TE polarization. In addition to their high absorption losses, however, the higher-order TE modes cannot be efficiently excited from a centrally aligned single-mode fiber due to the large mismatch of mode field sizes and the symmetry canceling. Therefore, the designed Ge nanostrip waveguide is expected to transmit only the TM fundamental mode in practice.

### 2.2. Fabrication Workflow

The fabrication of the Ge nanostrip follows a simple and standard process, as summarized in Figure 2a, taking advantage of the high flexibility and tolerance on the polymer waveguide platform. The ZPU450 under-cladding is first spin-coated and UV/thermally cured on a 4-inch silicon wafer. The nanostrips are defined by standard contact photolithography (MA6, SUSS MicroTec, Garching, Germany). After that, the Ge film is deposited by e-beam evaporation in high vacuum (<4.4 × 10^−5^ Pa) but at room temperature. A dummy silicon chip is also placed in the evaporation chamber for thickness control and ellipsometer measurement. The waveguide structures are transferred through a lift-off process. The upper-cladding is spin-coated and cured. Finally, the wafer is diced into bars/chips with a standard sawing machine. As thermal annealing is not performed, the deposited Ge layer should be amorphous. Figure 2b shows the detailed microscopic images of the 4 μm width Ge waveguides (top view), indicating smooth edges and homogeneous adhesion.

### 2.3. Experimental Setup

The photograph of the experimental setup is shown in Figure 3a. On the input side, light from a tuneable laser (T100S-HP, EXFO, Quebec City, QC, Canada) and a 2004 nm CW laser (Ultraline 2000, NPI Lasers, Nanjing, China) can be manually switched to couple into the waveguide. The fiber-chip-fiber alignment is facilitated by a set of fine positioning stages. The 3-axis piezoelectric controller can move the fiber at a precision of ±10 nm along X, Y, and Z directions. A pair of lensed fiber (TSMJ-X-1550-9/125-0.25-7-2.5-14-2-AR, OZ Optics, Ottawa, ON, Canada) is adopted for waveguide coupling for better efficiency, as shown in Figure 3b,c. On the output side, an optical component tester (CT440-PDL, EXFO, Quebec City, QC, Canada) is used for the collection and analysis of the transmitted light from 1500 nm to 1600 nm. For the 2004 nm laser, the output light is coupled to a thermopile optical power meter (3A, Ophir, Jerusalem, Israel) for characterization.

## 3. Results

### 3.1. Waveguide Measurement

A standard cut-back method measurement is performed using three sets of identical Ge nanostrip waveguides with different lengths, i.e., 0.5 cm, 1 cm, and 1.5 cm, respectively. First, the Ge waveguide chip is placed between the lensed fiber pair and finely aligned. The power from the tuneable laser is set to 10 mW. The fiber polarization controller is finely adjusted to make the transmitted power reach its minimal value at the detector. Since the Ge waveguide can transmit the TM mode with significantly lower loss, the input polarization state, giving the minimum transmitted power, is believed to be the TE polarization. After scanning the transmitted power under the current polarization state from 1500 nm to 1600 nm, the laser system automatically rotates the polarization direction by 90°, i.e., to the TM polarization, and repeats the wavelength scan. For the second measurement, the input is switched to the 2004 nm laser (also set to 10 mW output power), and the output fiber is connected to the thermopile power meter. The polarization adjustment is repeated, and the transmission values are obtained.

The fiber-to-fiber transmission without the waveguide chip in the middle serves as a reference for ruling out the system’s influence. Linear fitting of the cut-back measurement provides a clear indication of the waveguide propagation loss (slope) and the coupling loss with the fiber (intercept). The results for the 1500–1600 nm range are plotted in Figure 4a; however, only for the TM mode. For the TE polarization, we have only detected light power at noise level (−70 dBm) for the 0.5 cm long waveguide, and the loss is expected to be well over 100 dB/cm. As indicated by the gray curve in Figure 4a, the waveguide propagation loss gradually decreases from 20.56 dB/cm at 1500 nm to 11.75 dB/cm at 1600 nm. At 2004 nm, the transmitted power of the TE-polarized light input still stays below the detection limit (−30 dBm) of the chosen power meter, while the linear fitting reveals a propagation loss of 4.89 dB/cm for the TM mode, as shown in Figure 4b. The TE polarization undergoes a strong absorption and cannot be decisively measured, even at 2004 nm, while the TM polarization shows an increasing degree of transparency from 1500 nm to 1600 nm and further to 2004 nm. The comparison shown in Table 1 reveals a better transmission performance of the polymer-embedding Ge nanostrip waveguide compared to previous demonstrations on conventional material platforms (such as GeOI) at a ~2 μm wavelength.

On the other hand, the Ge nanostrip waveguide with a cross-section of 4 μm × ~40 nm is only demonstrated. The ~40 nm thickness is chosen since the waveguide principle is similar to the high-aspect-ratio dielectric waveguide, e.g., a silicon nitride waveguide in glass [38]. A thicker Ge layer will increase the propagation loss. Numerical simulations reveal that smaller Ge nanostrip widths/thicknesses can further weaken the mode confinement and effectively reduce the propagation loss of the TM fundamental mode. We project that the propagation loss will be lower than 2 dB/cm, including the absorption in the polymer cladding at a 2 μm wavelength.

Next, the ellipsometer measurement is performed on the dummy sample to obtain the optical parameters (*n* and *κ*). Considering the random atom orientations within the amorphous Ge film in the absence of long-range-order and the fact that a ~40 nm thick layer already stacks ~150 Ge atoms, the in-plane polarization (ordinary light or “o” light, referring to the TE polarization in the waveguide) and the vertical polarization (extraordinary light or “e” light, referring to the TM polarization in the waveguide) “sees” nearly the same optical constants. Therefore, an isotropic material model is adopted for the ellipsometry. The original data of the test sample are measured by 380–1650 nm. The refractive index *n* and the extinction coefficient *κ* are well-fitted and plotted as the solid curves in Figure 5. One Tauc–Lorentz oscillator and two Lorentz oscillators are applied to reshape the imaginary part of the complex dielectric function, i.e., the extinction coefficient *κ*, which ensures a low MSE while avoiding overfitting by excessive oscillators. Both optical constants exhibit similar trends to the previously published reference [40]. Since the light source of the ellipsometer cuts off at 1650 nm, the optical constants at longer wavelengths need to be extended by mathematical fitting, and the Cauchy model is considered a suitable candidate for such treatment [41].

The Cauchy model provides
(1)$$\begin{array}{c} n=A+B/\lambda^{2}+C/\lambda^{4} \end{array}$$
(2)$$\begin{array}{c} \kappa ={\;A}_{\kappa }e^{B_{\kappa }\left(1240/\lambda -1240/\lambda_{b}\right)} \end{array},$$
where *λ_b_* is usually a value near the shortest wavelength of the fitting region. Data in the 1400–1650 nm wavelength range in the original *n*-*κ* model are selected as the extension base. The dashed curves in Figure 5 plot the extension results. The mean square error (MSE) stays in the order of 1 × 10^−6^, indicating a reliable fitting for the model extension.

### 3.2. n-κ Model Correction

In ellipsometry, the imaginary part of the complex dielectric function is usually fitted by a combination of independent oscillators. For the measurement of thin films with strong optical absorption, the absorbing regions can be described by several symmetric/asymmetric oscillators, while their “tails” stack up in the near-transparent region. The consequence is that *κ* can only accumulate with the increasing number of oscillators, resulting in an overestimation of the extinction coefficients near the cut-off wavelength. Moreover, this accumulation is hard to tune down by modeling unless a simpler oscillator combination is adopted; however, this often results in a higher MSE. On the other hand, accurate optical parameters are crucial for the precise component design, e.g., a saturable absorber in a laser cavity. The existing Ge *n*-*κ* model, especially the extinction coefficient *κ*, needs confirmation and possibly a correction to match the experimental results.

In the first step, the real part (*n*) of the complex index is investigated. Since *n* is directly related to the eigenmode field size, the deviation of the model can be revealed by comparing the measured MFA with the simulated value using a commercial eigenmode solver. As shown in Figure 6a, an imaging system is established with a diaphragm, a set of lenses, a polarizer, and a camera. This setup is similar to a commercial beam profiler, which has been extensively used to examine the laser beam waist profile [42,43]. The tuneable laser from 1500 nm to 1600 nm is coupled to the chip, while the beam spot at the output facet is magnified on the camera through the imaging system. The polarizer direction is first adjusted as parallel to the in-plane polarization direction of the Ge waveguide. Then, the polarization controller is finely tuned to make the beam spot completely invisible on camera, reaching the TE mode. Subsequently, the polarizer is rotated by 90°, and the imaged mode field is considered as the pure TM mode. The first row in Figure 6b plots the captured spots from the waveguide output facet at the 1500 nm, 1550 nm, and 1600 nm wavelengths, respectively. The mode field of a standard single-mode fiber at the 1550 nm wavelength is also captured as a reference. The second row plots the simulated mode fields at corresponding wavelengths using the *n*-*κ* model presented in Figure 5 (for Ge waveguide modes). The MFA of the captured waveguide mode can be calculated as
(3)$${\mathrm{MFA}}_{\mathrm{WG}\;\mathrm{mode}}=\frac{{\mathrm{Effective}\;\mathrm{Pixel}}_{\mathrm{WG}\;\mathrm{mode}}\cdot {\mathrm{MFA}}_{\mathrm{fibre}\;\mathrm{mode}}}{{\mathrm{Effective}\;\mathrm{Pixel}}_{\mathrm{fibre}\;\mathrm{mode}}},$$
where the reference MFA*_fibre mode_* is ~85 μm^2^ at 1550 nm from the fiber data sheet, and effective pixel stands for the count of pixels in which the captured light power is no smaller than 1/*e*^2^ of the peak value. Figure 6c provides a comparison of the measured/simulated MFAs at 1500 nm, 1550 nm, and 1600 nm wavelengths, respectively. Each measurement is repeated multiple times, and the average value is taken. The deviations between measured/simulated MFAs stay within 5%.

In the second step, the imaginary part *κ* is corrected with the previously obtained transmission characteristics of the Ge waveguides. As listed in the first row of Table 2, six wavelengths with a step size of 20 nm are selected as samples. As presented in the second row, corresponding Ge core absorptions at these wavelengths are extracted from the gray curve (Ge waveguide loss vs. *λ*) in Figure 4a. According to the material datasheet, the polymer cladding exhibits an absorption loss of ~0.35 dB/cm at the 1550 nm wavelength. This loss is slightly wavelength-dependent from 1500 nm to 1600 nm but remains about two orders of magnitude smaller than the Ge material absorption. Therefore, we neglect the polymer absorption in the calculation. Since no etching is involved in the definition of the Ge nanostrip, the waveguide scattering loss can also be neglected. Therefore, the waveguide loss arises mainly from the Ge core absorption.

Next, we establish the Ge waveguide model in the commercial eigenmode solver and scan *κ* in the Ge material definition at all six sample wavelengths to make the simulated propagation loss as close as possible to the measured value. The given *n* of the Ge core for these simulations is listed in the third row of Table 2, which is directly obtained from the ellipsometer measurement in Figure 5. Finally, the optimized *κ* is listed in the fourth row of Table 2 and is considered to reveal the real *κ* at the corresponding wavelength. The relation of *κ* vs. *λ* from 1500 nm wavelength is fitted by the Cauchy model in Equation (2). The corrected *κ* is plotted as the dashed curve in Figure 7, which exhibits a significant offset from the original model. The MSE of the Cauchy fitting is in the order of 1 × 10^−8^.

The corrected *n*-*κ* model is further examined at the 2004 nm wavelength. First, a series of polymer channel waveguides with a 1.47–1.45 core-cladding index contrast is measured to estimate the polymer absorption at 2004 nm. The cross-sectional design of the waveguide core is 3.5 μm × 3.5 μm. As shown in Figure 8a, the cut-back fitted propagation loss of the TE and TM modes is 3.27 dB/cm and 3.16 dB/cm, respectively. The sidewall scattering loss of the polymer is estimated to be 0.14 dB/cm at the 2004 nm wavelength by Rayleigh scattering principles. Due to similar material compositions, the cladding polymer (ZPU450) and the core polymer (ZPU470) also share similar optical absorptions. Therefore, the polymer absorption is determined to be 3.02 dB/cm for the TM polarization at 2004 nm, as summarized in Figure 8b.

For the Ge waveguide, the dominating field component E_z_ of the TM fundamental mode is mainly concentrated in the polymer cladding. Thus, as plotted in Figure 8b, the absorption for the polymer cladding is already obtained as 3.02 dB/cm, and the absorption of the Ge core is calculated to be 1.87 dB/cm at the 2004 nm wavelength. Finally, the corrected material *n*-*κ* data are imported into the established Ge waveguide model in the commercial eigenmode solver. At the 2004 nm wavelength, the solver provides a propagation loss of 1.71 dB/cm (without polymer absorption). The 0.16 dB/cm deviation from the measured value may be attributed to the slight thickness deviation of the Ge layer in the experiment. Nevertheless, the results above demonstrate the accuracy and credibility of the correction method for the optical parameters of Ge. This waveguide-assisted characterization method can also be extended to other thin-film materials in the near-transparent region with evident bandgaps.

## 4. Discussion

The bulk crystalline germanium has a typical optical bandgap of 0.66 eV (an absorption cut-off wavelength of ~1850 nm). However, the experimental results shown in Figure 4 prove that the Ge nanostrip waveguide still absorbs above 2000 nm. On the other hand, the Tauc plot has become a simple but convenient tool for estimating the optical bandgap of amorphous thin semiconductors [44], i.e.,
(4)$$\begin{array}{c} \begin{array}{c} \sqrt{\alpha h\nu }\propto \left(h\nu -E_{g}\right) \end{array} \end{array},$$
where *α* is the absorption coefficient, *hν* is the photon energy, and *Eg* is the bandgap. The relation indicates that the intersection abscissa of the curve’s tangent and horizontal axis is the corresponding bandgap. As Figure 9 shows, the triangle markers provide a Tauc plot of the ~40 nm thick amorphous Ge film obtained from the corrected *κ* in Figure 7. Evidently, the Tauc model does not generally show a single linear trend but rather exhibits a pronounced curvature near the absorption band edge [45,46]. Therefore, linear fitting is performed at both high-energy and low-energy regions for the possible bandgap range. As displayed in Figure 9, the fitting predicts a bandgap of 0.408 eV from the high-energy fit and 0.588 eV from the low-energy fit. Compared with the *Eg* of crystalline Ge (0.66 eV), the bandgap shrinkage in the fitting results is consistent with the redshift of the cut-off wavelength in the waveguide experiments.

The phenomenon can be explained by the model-solid theory for semiconductor films [47]. The unique optical and electrical properties of *α*-Ge have been interpreted by the continuous random network (CRN) model [48]. However, contrary to the view that only a short-range order (SRO) exists in *α*-Ge, pieces of evidence have shown that the medium-range order (MRO), i.e., topologically crystalline grains, can also be embedded in the CRN [49]. Therefore, the structure of the amorphous Ge film can be defined as *A_x_B*_1−*x*_, where *A* is the topologically crystalline grains (MRO), and *B* is the continuous random network (SRO) [50]. Further, *x* determines the volume fraction of the MRO, which is highly related to the preparation technology/conditions of the thin film. For the thin Ge film deposited using e-beam evaporation, *x* will exceed a critical value of *a*% [51,52]. Meanwhile, the optical bandgap can be expressed in a range as [51]
(5)$${m\;\leq \;E}_{g}\left(A_{x}B_{1-x}\right)\;\leq \;0.66\;\left(x\;\geq \;a\%,\;d\;>\;\sim 10\;\mathrm{nm}\right),$$
where *m* is the limit of the narrowing bandgap, and *d* is the film thickness. With the increased *x* (MRO dominants) and the accumulation of film thickness, the strain exerted on topologically crystalline grains will build up. The strain causes the topologically crystalline grains to deform, which in turn shrinks the bandgap according to the model-solid theory [51]. Therefore, if the amorphous Ge core is further replaced by a crystalline nanostrip, this grain order effect can be significantly suppressed and result in a lower waveguide loss both for TE and TM polarization above the cut-off wavelength. This analysis may also inspire potential flexible bandgap manipulations through thickness, strain, curvature, and other physical parameters, offering more degrees of freedom for exploiting *α*-Ge films.

## 5. Conclusions

To summarize, a germanium nanostrip waveguide embedded in polymer has been fabricated using simple and low-cost technology. The Ge waveguide generates high polarization extinction and only supports effective transmission of the TM mode. From the cut-back measurement, the propagation loss of the TM mode exhibits a clear decline from 1500 nm to 2004 nm. The optical constants of an amorphous Ge layer, i.e., the refractive index *n* and extinction ratio *κ*, are measured and modeled by a spectrometric ellipsometer. Considering the ambiguous *n*-*κ* model by ellipsometry, verification and corrections are implemented based on the material model and the waveguide measurement regarding the mode field size and the propagation loss spectrum. It turns out that the real part of the refractive index is still accurate using the isotropic model in ellipsometry, while the extinction ratio must be suppressed to represent the experiment results with minimal MSE. The Tauc plot and fittings of the corrected *κ* indicate a possible bandgap range of 0.408–0.588 eV, showing a clear narrowing when compared to crystalline bulks (0.66 eV). This effect can be attributed to the grain group distribution in amorphous Ge films and can be further confirmed by waveguide transmission characterizations using laser sources at longer wavelengths.

To conclude, this work demonstrates a low-cost, easy-to-fabricate, and polarization-selective waveguide that can operate at NIR/MIR wavelengths. Moreover, an innovative method for correcting the optical parameters of thin-film materials is also proposed by the transmission characteristics in the guided-wave domain. We believe this work will prove useful in the application of light-absorbing films as key functional layers on the polymer photonic integration platform. In particular, such a heterogenous waveguide would allow flexible design and low-cost fabrication of the critical saturable absorber component in the external cavity of a hybrid laser for mode-locking operation.

## Figures and Tables

**Figure 1 polymers-15-04093-f001:**
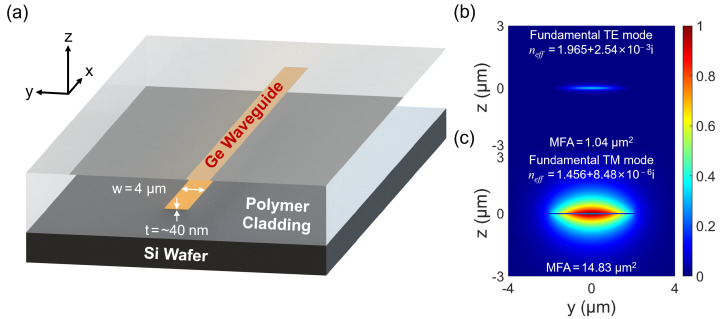
(**a**) A 3D schematic view of the germanium nanostrip waveguide in polymer. The width and thickness of the Ge core are 4 μm and ~40 nm, respectively. (**b**,**c**) Normalized mode profiles of the TE and TM fundamental modes after a propagation distance of 50 μm at 1550 nm wavelength. (MFA: mode field area).

**Figure 2 polymers-15-04093-f002:**
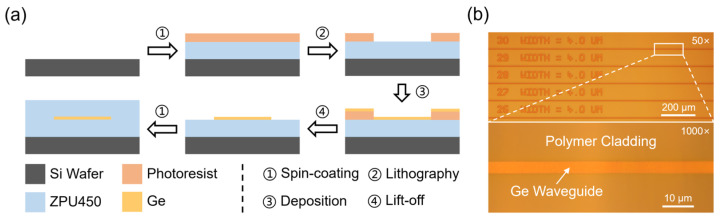
(**a**) Flowchart of the Ge waveguide fabrication process. (**b**) Microscopic images of the 4 μm wide Ge waveguides (top view).

**Figure 3 polymers-15-04093-f003:**
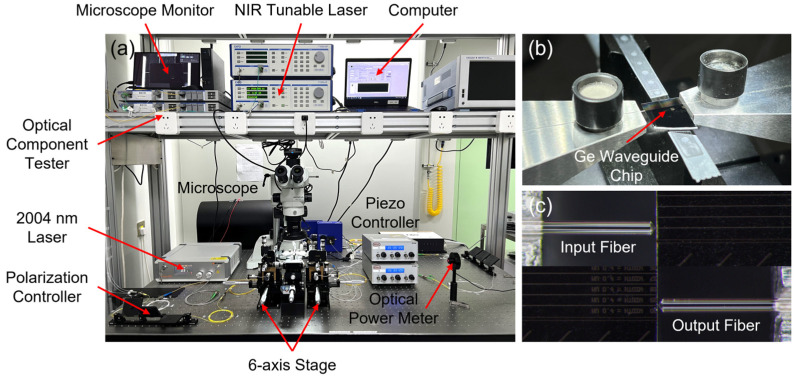
(**a**) Photograph of the experimental setup. (**b**) Zoomed-in view of the Ge waveguide chip. (**c**) Microscopic images showing the fiber-waveguide-fiber coupling.

**Figure 4 polymers-15-04093-f004:**
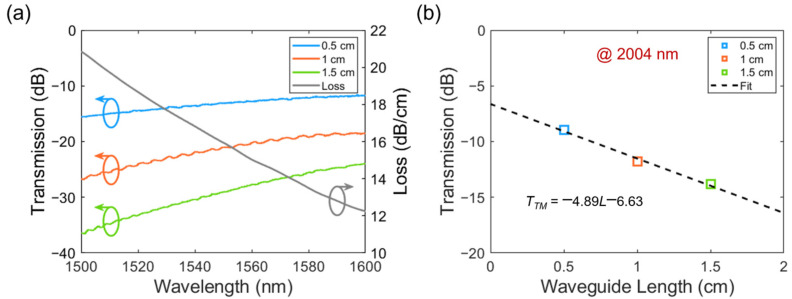
(**a**) Measured transmission (TM polarization) of the 0.5 cm, 1 cm, and 1.5 cm long Ge strip waveguides from 1500 nm to 1600 nm. The relation of the waveguide loss vs. wavelength is obtained by the cut-back method with 1 nm wavelength step. (**b**) Measured transmission (TM polarization) of 0.5 cm, 1 cm, and 1.5 cm long Ge strip waveguides at 2004 nm wavelength. The propagation loss of the waveguide is fitted as 4.89 dB/cm.

**Figure 5 polymers-15-04093-f005:**
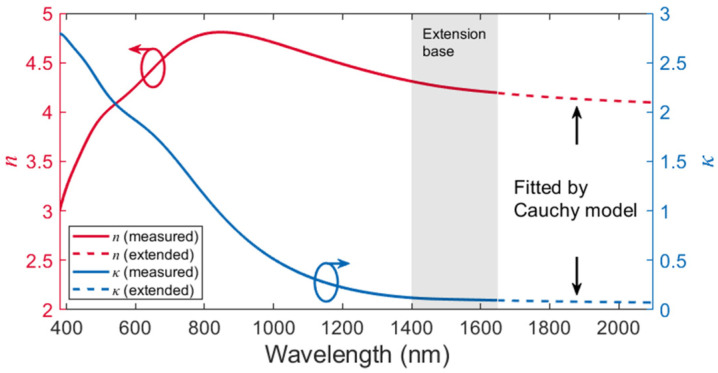
The *n*-*κ* model of the ~40 nm thick amorphous Ge film by spectroscopic ellipsometry.

**Figure 6 polymers-15-04093-f006:**
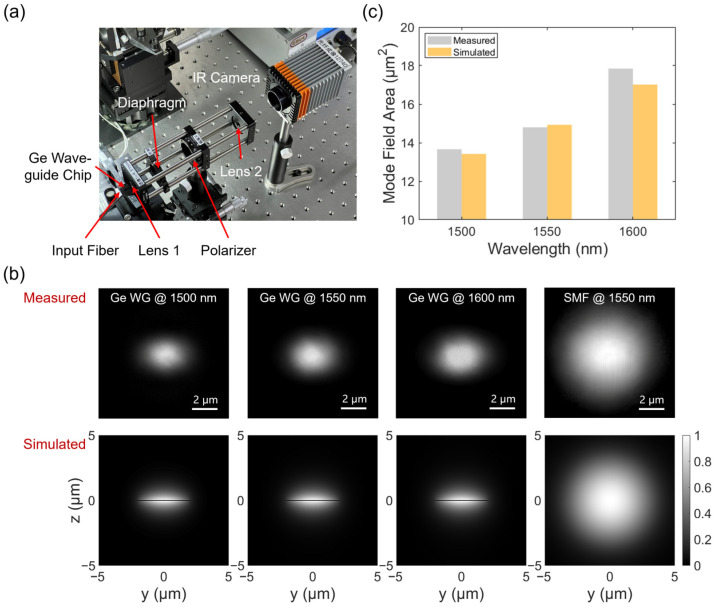
(**a**) Photograph of the imaging system. (**b**) Row 1: captured spots (TM mode) from the output facet of the Ge nanostrip waveguide at 1500 nm, 1550 nm, and 1600 nm wavelengths, and the spot from a reference single-mode fiber at 1550 nm wavelength, respectively. Row 2: simulated profiles (using the *n*-*κ* model in Figure 5) of the Ge waveguide (TM fundamental mode) and the reference single-mode fiber. (**c**) Comparison between measured and simulated MFAs of the Ge waveguide at 1500 nm, 1550 nm, and 1600 nm wavelengths, respectively.

**Figure 7 polymers-15-04093-f007:**
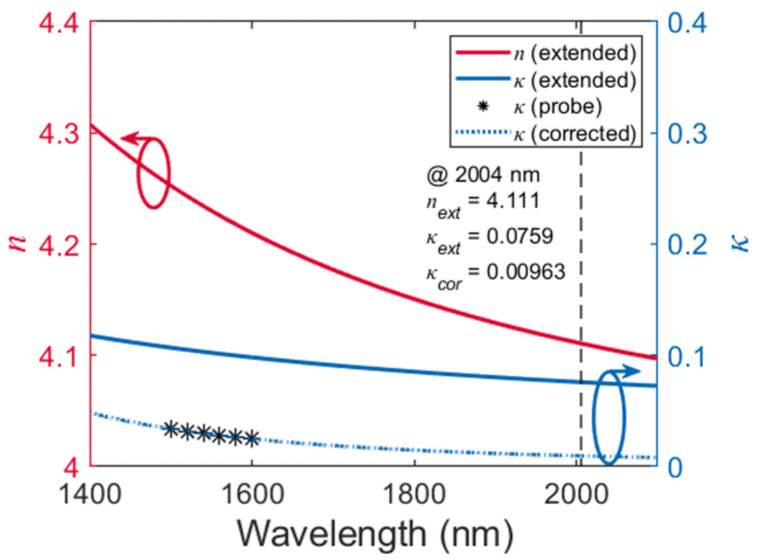
Comparison between the original and corrected *n*-*κ* model of the ~40 nm thick amorphous Ge film. Asterisks represent the values of “probe *κ*”.

**Figure 8 polymers-15-04093-f008:**
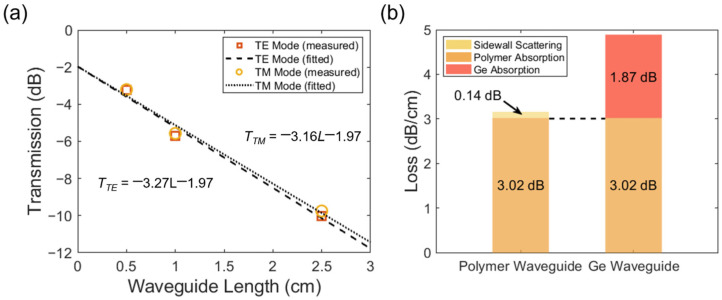
(**a**) Measured transmission of 0.5 cm, 1 cm, and 2.5 cm long polymer waveguides at 2004 nm for both TE and TM fundamental modes. The propagation loss of the TE/TM fundamental modes is fitted as 3.27 dB/cm and 3.16 dB/cm, respectively. (**b**) Loss compositions of the polymer waveguide and the Ge nanostrip waveguide at 2004 nm.

**Figure 9 polymers-15-04093-f009:**
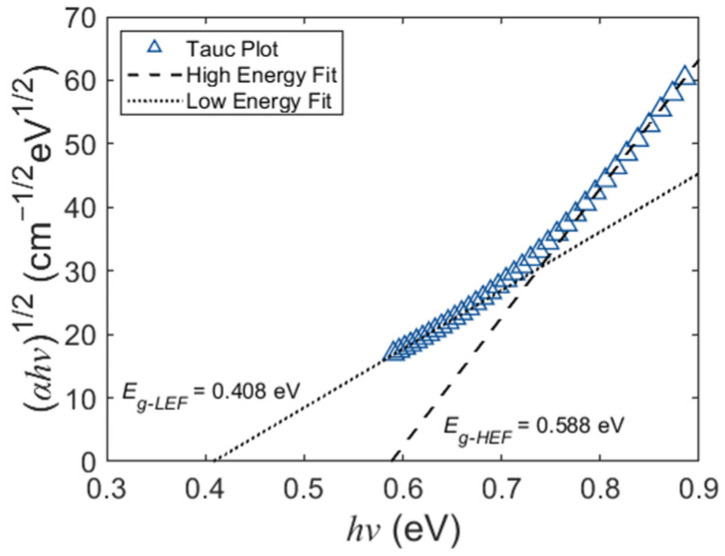
Representative Tauc plots and corresponding linear fittings of the ~40 nm thick amorphous Ge film after optical parameter correction.

**Table 1 polymers-15-04093-t001:** Comparison of Ge waveguides propagation loss at 2 μm wavelength.

Ref. No.	Material Platform	Cross-Sectional Parameter	Waveguide Type	Propagation Loss (dB/cm)
[13] (Sample A)	GeOI	1 μm (w) × 0.3 μm (h)	Rib	25.2
[13] (Sample B)	GeOI	1 μm (w) × 0.3 μm (h)	Rib	10.7
[14]	GeOI	1 μm (w) × 0.27 μm (h)	Strip	84
[19]	GeOI	2 μm (w) × 0.3 μm (h)	Rib	14
This work	Polymer	4 μm (w) × ~40 nm (h)	Strip	4.89

**Table 2 polymers-15-04093-t002:** Parameters for *n*-*κ* model correction.

**Correction *λ* (nm)**	1500	1520	1540	1560	1580	1600
**Ge core absorption (dB/cm)**	20.56	18.55	16.50	14.57	12.93	11.75
**Measured *n* by ellipsometer**	4.251	4.241	4.232	4.225	4.218	4.210
**Probe *κ***	3.39 × 10^−2^	3.17 × 10^−2^	3.00 × 10^−2^	2.79 × 10^−2^	2.62 × 10^−2^	2.49 × 10^−2^
**Fitted relation of *κ* vs. *λ*** **by Cauchy model**	$\kappa =0.0339e^{6.056(1240/\lambda -\;0.827)}$					

## Data Availability

The data presented in this study are available from the corresponding author upon reasonable request.

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
