# Peer review of "Polymer-Embedding Germanium Nanostrip Waveguide of High Polarization Extinction"

_polymers, 2023, doi:10.3390/polym15204093_

Round 1

Reviewer 1 Report

The manuscript "Polymer-Embedding Germanium Nanostrip Waveguide of High Polarization Extinction" show an interesting study of Ge nanostrips embedded in polymers and their Polarization effect. There are some parts that need to be addressed.

1. The authors used a lot of abbreviation as example TE and TM not explained throughout the manuscript. Please explain it where mention first as the abstract. Others are there as well that needs more definition as example Im(ε) or n-κ Model. Please check also other abbreviations throughout the manuscript.

2. Why the author used perfluorinated acrylate for a polymer why not silicon based as most research direction goes that direction? What advantages have Exguide ZPU12-450? Please give more explanation in the introduction.

3. There is some figure as example Figure 3 that belongs to experimental part and not result hence its just the measurement device you using and there nothing novel in that.

4. For the results obtained in this research there should be a proper comparison to other research made using as well Ge in polymers. Please include a Table at best place in the discussion part to show where your results are more favorite than others.

Reviewer 2 Report

The manuscript is a comprehensive experimental and theoretical study of the waveguide properties of Ge nanostrip in a polymer core in the TM mode. The arguments are considered convincing, the article is well written, well illustrated and contains interesting results.

The work could be improved if the following issue was discussed:

If we replace the amorphous Ge core with a crystalline nanostrip, how can this affect the operation of the waveguide?
